# End-stage renal disease accompanied by mild cognitive impairment: A study and analysis of trimodal brain network fusion

Jie Chen[1], Tongqiang Liu[2], Haifeng Shi[3]*

1 Department of Security, Huaide College of Changzhou University, Jingjiang, Jiangsu, China, 2 Department of Nephrology, The Affiliated Changzhou No.2 People's Hospital of Nanjing Medical University, Changzhou, Jiangsu, China, 3 Department of Radiology, The Affiliated Changzhou No.2 People's Hospital of Nanjing Medical University, Changzhou, Jiangsu, China

ʘ These authors contributed equally to this work.
* doctorstone771@163.com

**Data Availability Statement:** Data cannot be shared publicly because of legal or ethical grounds. Data are available from the Ethics Committee of Changzhou Second People's Hospital affiliated to Nanjing Medical University for researchers who

## Abstract

The function and structure of brain networks (BN) may undergo changes in patients with end-stage renal disease (ESRD), particularly in those accompanied by mild cognitive impairment (ESRDaMCI). Many existing methods for fusing BN focus on extracting interaction features between pairs of network nodes from each mode and combining them. This approach overlooks the correlation between different modal features during feature extraction and the potentially valuable information that may exist between more than two brain regions. To address this issue, we propose a model using a multi-head self-attention mechanism to fuse brain functional networks, white matter structural networks, and gray matter structural networks, which results in the construction of brain fusion networks (FBN). Initially, three networks are constructed: the brain function network, the white matter structure network, and the individual-based gray matter structure network. The multi-head self-attention mechanism is then applied to fuse the three types of networks, generating attention weights that are transformed into an optimized model. The optimized model introduces hypergraph popular regular term and $L_1$ norm regular term, leading to the formation of FBN. Finally, FBN is employed in the diagnosis and prediction of ESRDaMCI to evaluate its classification performance and investigate the correlation between discriminative brain regions and cognitive dysfunction. Experimental results demonstrate that the optimal classification accuracy achieved is 92.80%, which is at least 3.63% higher than the accuracy attained using other methods. This outcome confirms the effectiveness of our proposed method. Additionally, the identification of brain regions significantly associated with scores on the Montreal cognitive assessment scale may shed light on the underlying pathogenesis of ESRDaMCI.

## 1 Introduction

End-stage renal disease (ESRD) refers to the final stage of chronic kidney disease resulting from various health issues [1]. Cognitive dysfunction, on the other hand, encompasses

meet the criteria for access to confidential data. The data underlying the results presented in the study are available from the Affiliated Changzhou No.2 People's Hospital of Nanjing Medical University.E-mail: Liuyf1106@126.com Contact information for the Ethics Committee: 2392058632@qq.com/15804942480.

**Funding:** This work was supported by grants awarded to HS by Top Talent of Changzhou "The 14th Five-Year Plan" High-Level Health Talents Training Project (grant No. 2022CZBJ072), Changzhou Sci&Tech Program (grant No. CE20235062) and Clinical Research Project of Changzhou Medical Center of Nanjing Medical University (grant No. CMCC202306).

**Competing interests:** The authors have declared that no competing interests exist.

abnormalities in advanced cognitive processes within the brain [2]. As the population ages and chronic diseases become more prevalent, cognitive dysfunction in ESRD patients is emerging as a significant concern. In some studies, this percentage is believed to be even higher, reaching up to two-thirds of all patients [3]. The occurrence of cognitive dysfunction in ESRD patients severely impacts their daily lives, educational pursuits, professional endeavors, and social activities. This places a substantial burden not only on the individuals themselves but also on their families and society as a whole [4]. Consequently, early detection and intervention are of utmost importance. Mild cognitive impairment (MCI) represents the intermediate stage between the normal decline in memory and thinking associated with aging and the more pronounced cognitive deterioration observed in dementia. With appropriate cognitive training and rehabilitative treatments, the progression of cognitive dysfunction in MCI patients can be delayed, and in some cases, even reversed, leading to a return to a normal state [5]. Nevertheless, the precise neuropathological mechanisms underlying ESRD accompanied by mild cognitive impairment (ESRDaMCI) remain unclear, impeding the development of effective treatment strategies.

In recent years, there has been significant progress in the development of multimodal neuroimaging techniques for studying brain diseases, including ESRD-related neurological complications. These techniques include functional magnetic resonance imaging (fMRI) [6], structural magnetic resonance imaging [7], magnetic resonance spectroscopy [8], fluorodeoxyglucose positron emission tomography [9], single photon emission tomography [10], arterial spin labeling [11], diffusion tensor imaging [12], and diffusion kurtosis imaging (DKI) [13]. DKI has gained rapid development and has provided valuable insights into the pathophysiological mechanisms of brain diseases from various perspectives such as deposition of pathological markers, brain structural connectivity, and functional connectivity. It serves as a non-invasive tool for exploring the brain and its connectivity patterns, revealing previously unknown brain functions and structures. DKI is based on a non-Gaussian diffusion model that considers changes in kurtosis, which is a measure of the pattern of distribution in probability theory and statistics [14]. DKI overcomes the limitations of diffusion tensor imaging and provides crucial microstructure information about the brain, particularly gray matter, while also supplementing white matter information. The DKI index can enhance the sensitivity and specificity of disease assessment and serve as an important imaging parameter for evaluating disease progression and treatment response. Research has indicated that DKI is more sensitive than diffusion tensor imaging in detecting changes in brain tissue microstructure. An increase in diffusion kurtosis value (KV) is generally associated with an increase in microstructure complexity [15].

The analysis of brain networks (BN) in neuroimaging data has proven to be a valuable approach for uncovering the structure and function of the brain [16]. BN offers a powerful representation of the patterns of interaction between different regions of the brain. In a BN model, brain regions are represented as nodes based on a physiological template, and the connections between these nodes are derived from non-invasive imaging techniques, capturing regional interactions. Investigating the relationship between changes in brain functional networks and MCI in patients with ESRD is an area of active research interest [17]. Functional networks provide insights into the patterns of functional connectivity between various brain regions. In recent years, there have been studies investigating the BN features of ESRDaMCI using brain imaging techniques, as demonstrated by Xi et al. [18]. However, the specific mechanisms and contributing factors underlying these network abnormalities remain controversial. Conversely, reports on brain structural network studies in ESRDaMCI are scarce. The structural network of the brain provides insights into its physical foundation and connectivity. Exploring changes in structural networks in ESRD patients and their association with MCI is

essential for a deeper understanding of ESRDaMCI. In addition, there have been limited studies on multimodal fusion research methods integrating functional and structural networks in ESRDaMCI patients. Xi et al. [19] employed a hypergraph-based approach to fuse functional and white matter structural networks in ESRDaMCI patients, yielding favorable classification results and identifying discriminative brain regions. The utilization of multimodal fusion methods can offer more comprehensive and accurate brain information, revealing correlations between different modes of BN. Therefore, the application of multimodal fusion in investigating ESRDaMCI provides a novel perspective and enables in-depth analysis of the pathophysiological mechanisms involved.

In recent years, various machine learning techniques have been proposed and applied to fuse multiple modes in the analysis of BN. Some approaches treat functional and structural networks of the brain as independent feature extractions. For instance, Dyrba et al. [20] combined functional and structural networks using multi-core support vector machines (SVM) for Alzheimer's disease identification. Liu et al. [21] proposed a multi-view embedding method for BN cluster analysis, representing functional and structural networks as multi-view data. However, these methods overlook the underlying relationship between brain activity and physical connectivity among brain regions. Another approach involves bootstrapping one network construction with another network. Chu et al. [22] defined seed and target regions anatomically using fMRI data and constructed a structural network based on high-angular-resolution diffusion imaging. Pinotsis et al. [23] constructed a structural network based on the relationship between theoretical graph properties and simulated functional dynamics. Nevertheless, these methods only consider direct connections in BN and do not account for indirect connections on a larger scale. In reality, brain regions often interact directly with several neighboring brain regions, forming complex interaction relationships. Ji et al. [24] employed hypergraphs to capture higher-order relationships among multiple nodes, utilizing them in the study of BN. More recently, attention mechanisms have been employed to fuse multimodal BN, allowing for a broader consideration of node interactions across multiple modes. For example, Huang et al. [25] integrated brain functional and white matter structural networks by combining single-headed attention mechanisms with the diffusion process, resulting in promising outcomes.

Most fusion methods for BN extract interaction features between pairs of network nodes within each mode separately. They then splice and combine multiple sets of feature vectors, overlooking the correlation between different modal features and potentially useful information that may exist between two or more brain regions during feature extraction. This information is crucial for understanding the pathological basis of ESRDaMCI [26]. To address these issues, this study proposes a model utilizing a multi-head self-attention mechanism to fuse brain functional networks, white matter structural networks, and gray matter structural networks in order to construct brain fusion networks (FBN) that capture the relationship between ESRDaMCI's cognitive function and network connectivity features. Specifically, the brain functional network is first constructed using time series data extracted from fMRI. The white matter structural network is constructed using the number of fiber bundles (FN) extracted from DKI. Additionally, the individual-based gray matter structural network is constructed using KV extracted from DKI. Based on these networks, the multi-head self-attention mechanism is employed to jointly learn the three modes of networks and train the attention weights. Subsequently, the value vectors are combined to obtain the feature connections, which are then transformed into an optimization model. To generate FBN, a hypergraph is constructed according to the obtained feature connections. The optimization model incorporates the hypergraph manifold regularization term (HMR) and $L_1$ norm regularization term. Finally, FBN is applied in the diagnosis and prediction of ESRDaMCI to evaluate its

classification performance. Correlation studies are also conducted to examine discriminative brain regions and cognitive dysfunction.

This study introduces several novel innovations and contributions: a) For the first time, an individual-based gray matter network specific to ESRDaMCI was constructed using KV extracted from DKI; b) The application of a multi-head self-attention mechanism is a novel approach to model ESRDaMCI's brain function network, as well as the fusion network of white matter structure and gray matter structure. This approach yielded impressive classification results; c) By utilizing the aforementioned fusion network, this study successfully identifies discriminative brain regions associated with the pathogenesis of ESRDaMCI. Furthermore, it analyzes the changes in topological attributes of these discriminative brain regions and identifies brain regions that significantly contribute to cognitive dysfunction.

## 2 Materials and methods

### 2.1 Data acquisition

Fifty-one ESRD patients diagnosed with MCI were enrolled from the Changzhou Second People's Hospital of Nanjing Medical University between April 20, 2021 and June 15, 2023, forming the ESRDaMCI group. The average age of the participants in this group was 50.05 ± 7.86 years, with an average length of schooling of 11.25 ± 3.15 years. All participants were right-handed and had no history of infectious diseases, cardiac cerebrovascular diseases, diabetes, or neurological disorders. Additionally, there were no contraindications to undergo a magnetic resonance imaging scan. In parallel, a control group comprising 39 healthy volunteers (Normal group) was recruited, with an average age of 48.37 ± 6.59 years and an average length of schooling of 9.73 ± 3.85 years. All participants in the Normal group were right-handed, in good health, and had no contraindications for magnetic resonance imaging scanning. The study protocol was reviewed and approved by the Ethics Committee of Changzhou Second People's Hospital affiliated to Nanjing Medical University, with the approval number KY032-01. Written informed consent was obtained from all participants prior to their inclusion in the study.

Prior to the image scan, all participants completed the Montreal Cognitive Assessment (MoCA) [27], a widely-used tool for assessing cognitive dysfunction and detecting various cognitive domains. It has been observed that the MoCA scale has a higher completion rate and sensitivity in identifying MCI in memory clinics, according to statistics from Changzhou Second People's Hospital Affiliated to Nanjing Medical University. Therefore, we utilized the MoCA scale to assess the cognitive function of both ESRD patients and normal individuals in this study. The MoCA scale comprises a total score of 30, with scores equal to or greater than 26 considered normal, scores between 18 and 26 indicating mild impairment, scores between 10 and 17 representing moderate impairment, and scores below 10 indicating severe cognitive impairment. To account for the potential impact of education on the results, one point was added to the total score for individuals with less than 12 years of schooling. Before the magnetic resonance imaging scan, a neurologist with 20 years of experience evaluated all participants' neuropsychological test results. The ESRD participants diagnosed with MCI had an average MoCA score of 21.30 ± 2.75. Detailed demographic information can be found in Table 1, a significance level of $P < 0.05$ was used to determine statistical significance.

### 2.2 Data processing

DKI preprocessing was conducted using the FSL (https://fsl.fmrib.ox.ac.uk/fsl/fslwiki/) and PANDA (http://www.nitrc.org/projects/panda) toolkits [28]. The following steps were performed: (1) Format conversion; (2) Eddy head motion correction; (3) Gradient correction and noise

**Table 1. Demographic information.**

| Items | ESRDaMCI group | Normal group | $t/\chi^2$ | $P$ |
|---|---|---|---|---|
| Age | 50.05±7.86 | 48.37±6.59 | 1.079 | 0.251 |
| Gender (M/F) | 24/27 | 24/15 | 0.341 | 0.536 |
| Education years | 11.25±3.15 | 9.73±3.85 | 0.973 | 0.771 |
| MoCA score | 21.30±2.75 | 27.27±1.24 | -13.728 | 0.000 |

reduction; (4) Mask image generation; (5) Parameter calculation; (6) Resampling; (7) Gaussian smoothing and registration; (8) White matter fiber bundle tracking (A deterministic fiber bundle tracking algorithm was employed, with tracking parameters set as anisotropy index < 0.15 and tracking angle < 45° to calculate indicators such as FN between brain regions [29]).

fMRI preprocessing was performed using SPM8 (http://www.fil.ion.ucl.ac.uk/spm/) and DPARSF (http://rfmri.org/dparsf) toolkits [30]. The following steps were executed: (1) Format conversion; (2) Slice timing correction; (3) Head motion realignment and normalization; (4) Smoothing; (5) Detrending; (6) Filtering; (7) Regression analysis; (8) Time series extraction.

## 2.3 Brain network construction

The selection of nodes in a BN depends on the spatial scale being investigated, which can be categorized into microscale, intermediate scale, and large scale. The microscale refers to the smallest research units in the brain, such as voxels and neurons. At the intermediate scale, the focus is on voxel clusters or neuron clusters within the brain. Large-scale studies aim to understand the connections and interactions between brain regions, often utilizing pre-defined brain partitions as templates. One widely used template is the AAL template, which provides a digital map of brain structures. In neuroimaging studies, the AAL template is commonly employed to label the neuroanatomical locations of brain function measurements in three-dimensional space [31]. In our study, we specifically investigate large-scale problems and adopt the AAL template to divide the brain into 90 regions, which will serve as nodes in our network. The choice of nodes in a BN may also vary depending on the specific research question being addressed. For our fMRI-based analysis, we define the edges of the brain functional network by computing Pearson correlation (PC) coefficients between the time series of different brain regions. Among the various parameters of DKI, the FN metric is known for its sensitivity to white matter, while the KV index is highly sensitive to gray matter [32]. Therefore, we utilize the PC coefficient of FN to construct the brain's white matter structure network, and compute the Euclidean distance of the KV metric to establish the brain's gray matter structure network at an individual level. Finally, we employ a multi-head self-attention mechanism to construct the FBN, and the construction process is depicted in Fig 1.

**2.3.1 Functional network.** To construct brain functional networks, the average time series signal of all voxels within a brain region is computed to represent the time series signal of that specific region. Various methods can be used to calculate correlation coefficients, such as PC coefficient and partial correlation coefficient. It has been noted in previous studies that the analysis of partial correlation coefficient is suitable when the sample size is larger than the number of brain regions. However, this method tends to underestimate various topological measures [33]. Therefore, in our study, we employ the PC coefficient to measure the strength of connections by calculating the PC coefficient between the time series of each brain region in each subject. The result is a symmetrical matrix representing the brain's functional network for each individual.

$$\boldsymbol{F}(i,j) = corr(x_i, x_j) \tag{1}$$

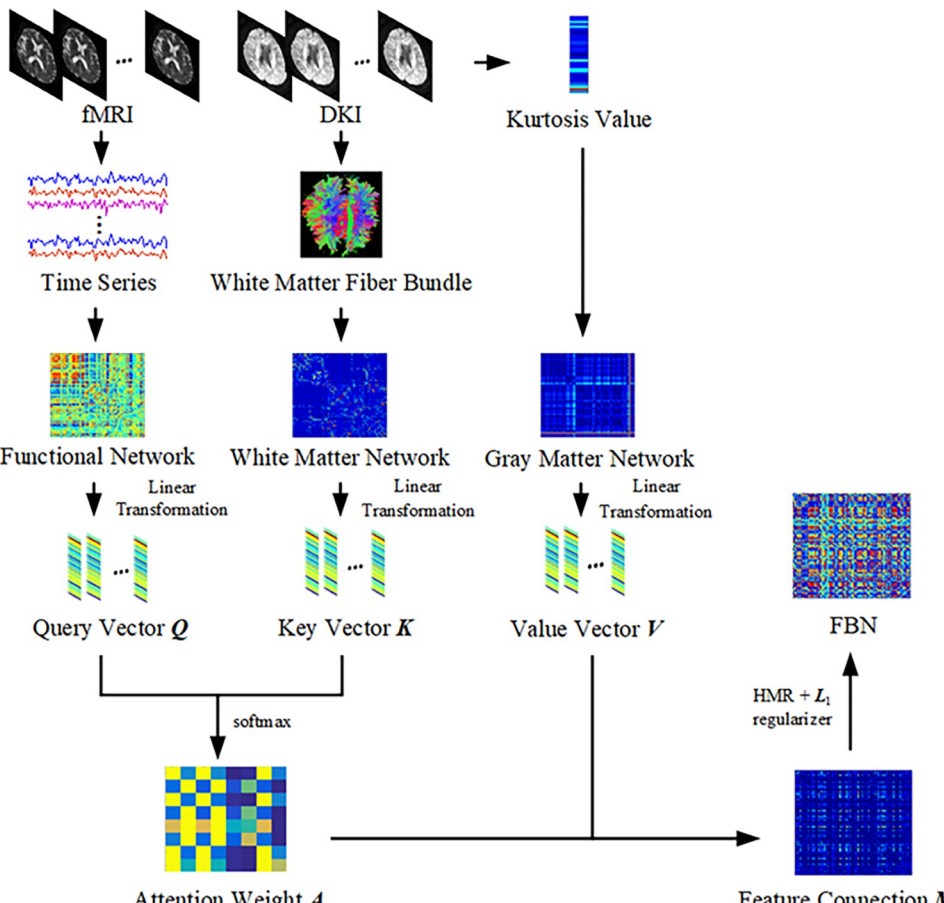

**Fig 1. Brain fusion network construction framework.** First, the brain function network was constructed using the time series extracted from fMRI, the brain white matter structure network was constructed using FN extracted from DKI, and the individual-based brain gray matter structure network was constructed using KV extracted from DKI. On this basis, the multi-head self-attention mechanism is used to learn the three modes of network interactively, and the attention weight is trained. Then, the value vector is combined to obtain the feature connections and converted into the optimization model. According to the generated feature connections, the hypergraph is constructed, and the HMR and $L_1$ norm regular terms are introduced into the optimization model to obtain FBN.

where $F$ represents the brain function network matrix, $x_i$ and $x_j$ denote the average time series of brain regions $i$ and $j$, respectively, and $corr()$ refers to the PC function.

**2.3.2 White matter structure network.** FN in the brain's white matter structural network provides an indication of the strength of its structural connections. Greater FN values between brain regions suggest a higher level of connection tightness. Specifically, a value exceeding 3 between two nodes implies the existence of a connection [34]. To obtain the FN values, we utilized the FACT tracking algorithm [35] in deterministic fiber bundle tracking based on DKI. Subsequently, the weighted brain white matter structure network was constructed using the PC method.

$$Fib(i,j) = count(fib_i, fib_j) \qquad (2)$$

$$B(i,j) = corr(Fib_i, Fib_j) \qquad (3)$$

where $Fib$ represents the FN value that connects brain region $i$ and $j$. The $count()$ function

calculates the number of FN values that satisfy the specified condition. The variables $fib_i$ and $fib_j$ correspond to the starting and terminating brain regions of a fiber bundle, respectively. Eq (2) describes a process wherein the entire set of fiber bundles is iterated, and for each bundle, its starting and ending brain regions are checked. If the starting point falls within brain region $i$ and the ending point falls within brain region $j$, the corresponding ***Fib***$(i, j)$ value is incremented by 1. The matrix ***B*** represents the brain white matter structure network, whereas $Fib_i$ and $Fib_j$ denote the FN values associated with brain regions $i$ and $j$, respectively.

**2.3.3 Gray matter structure network.** In the process of constructing the brain gray matter structure network, the KV of each subject's brain region was initially computed [36]. The preprocessed DKI data and AAL template data were read using the *niftiread* function and transformed into matrix format. To segment the gray matter region, a mask based on the AAL template was generated. By traversing each brain region defined by the AAL template data, the corresponding DKI data for that specific brain region was obtained using the mask. Subsequently, for each brain region, the KV was calculated using the *kurtosis* function, which relies on the fourth moment of the DKI data. This calculation provides quantitative information about the degree of spikiness or flattening in the data, aiding in understanding the shape characteristics of the gray matter structural network within the dataset [37]. The KV can be mathematically expressed as follows:

$$Kv(i) = (1/N)*sum((\boldsymbol{x}_i - mean(\boldsymbol{x}_i))^4)/(std(\boldsymbol{x}_i)^4) - 3 \qquad (4)$$

where $Kv$ represents an n-dimensional KV vector, $x_i$ corresponds to the DKI data matrix obtained for brain region $i$ using the provided mask. $N$ denotes the total number of brain regions. The function *mean*() calculates the mean value, while *std*() determines the standard deviation. A positive $Kv$ value indicates a relatively sharp (peak) distribution of data with a thicker tail. Conversely, a negative $Kv$ value suggests a relatively flat data distribution with a shorter tail. KV serves as a descriptor for the non-Gaussian diffusion of water molecules within the specific brain region.

To quantify the connections between the networks of gray matter structures in the brain, the Euclidean distance of the KV vector was computed. As the individual-based KV is an $N$-dimensional vector, traditional correlation measures such as PC cannot be directly applied to evaluate the correlation between one-dimensional vectors. Instead, the Euclidean distance can serve as a suitable method for assessing the correlation between KV vectors when constructing a weighted brain gray matter structure network [38]. Hence, we employed the Euclidean distance to calculate the degree of differentiation among individual KV vectors, enabling us to define the strength of connections between brain regions for each individual. This process generated an individual-based $N{\times}N$ matrix representing the gray matter structure network.

$$\boldsymbol{H}(i, j) = sqrt(sum((Kv_i - Kv_j)^2)) \qquad (5)$$

where *sum*() denotes the summation of the calculated elements, *sqrt*() represents the square root of the calculated elements, ***H*** represents the brain gray matter structure network, $Kv_i$ corresponds to the KV value in brain region $i$, and $Kv_j$ corresponds to the KV value in brain region $j$.

**2.3.4 Fusion network.** We employ a multi-head self-attention mechanism to combine the brain's functional network ***F***, white matter structure network ***B***, and gray matter structure network ***H*** into a unified network. This mechanism is suitable for scenarios involving intricate and nonlinear dependencies among different modalities, thereby enhancing the performance and representational capacity of the fused network [39]. By quantifying the interconnectedness between input features of each brain region and features from other modes, we adaptively

integrate diverse modal BN using the self-attention mechanism. Consequently, the characteristics of each modal BN can be influenced by the features of other modal BN, resulting in a fused network that incorporates cross-modal information. The model can be expressed as follows:

Assuming the number of brain regions as $N$ and the dimension of each network as $D$, the functional network $\boldsymbol{F} \in \boldsymbol{R}^{N \times N}$ can be denoted as $\boldsymbol{F} = [F_1, F_2, \ldots, F_N]$. Here, $F_i$ represents a $D$-dimensional vector that encapsulates the functional features of the $i$-th brain region. Similarly, the white matter structure network $\boldsymbol{B} \in \boldsymbol{R}^{N \times N}$ can be represented as $\boldsymbol{B} = [B_1, B_2, \ldots, B_N]$, with $B_i$ representing a $D$-dimensional vector representing the white matter structure features of the $i$-th brain region. Additionally, the gray matter structure network $\boldsymbol{H} \in \boldsymbol{R}^{N \times N}$ can be expressed as $\boldsymbol{H} = [H_1, H_2, \ldots, H_N]$, where $H_i$ corresponds to a $D$-dimensional vector capturing the gray matter structure features of the $i$-th brain region.

We propose a multi-head attention mechanism to capture associations between different networks. We define the number of heads as $X = 1, 2, \ldots x$, with each head having a dimension $D_x$. Within each head, the attention weights between brain region $i$ and region $j$ (where $j \neq i$) are learned using the attention mechanism. By employing linear transformations on the input data, they can be mapped to a feature space more suitable for attention computation. This approach enables interaction and fusion between different inputs through weighted fusion operations. Specifically, a linear transformation is applied to map the functional network $\boldsymbol{F}$ to a query vector $\boldsymbol{Q}$, the white matter network $\boldsymbol{B}$ to a key vector $\boldsymbol{K}$, and the gray matter network $\boldsymbol{H}$ to a value vector $\boldsymbol{V}$.

$$Q_i = \boldsymbol{W_Q} F_i \tag{6}$$

$$K_j = \boldsymbol{W_K} B_j \tag{7}$$

$$V_j = \boldsymbol{W_V} H_j \tag{8}$$

where $\boldsymbol{W_Q} \in \boldsymbol{R}^{D \times D_x}$, $\boldsymbol{W_K} \in \boldsymbol{R}^{D \times D_x}$, and $\boldsymbol{W_V} \in \boldsymbol{R}^{D \times D_x}$ are linear projection matrices that are obtained through training and learning.

Subsequently, the attention weight is calculated and normalized using the *softmax* function to ensure that the sum of the weights equals 1.

$$a_{ij} = softmax\left(\frac{Q_i K_j^{\mathrm{T}}}{\sqrt{D_x}}\right) \tag{9}$$

where $a_{ij} \in \boldsymbol{A}$ represents the attention weight between brain region $i$ and $j$, divided by $\sqrt{D_x}$ for scaling to balance the attention distribution. The self-attention output of the $i$ brain region is calculated by taking the weighted sum of the attention weight $\boldsymbol{A}$ and the value vector $\boldsymbol{V}$.

$$M_i = \sum_{j=1}^{N} (a_{ij} V_j) \tag{10}$$

Following the multi-head self-attention mechanism, we obtain $\boldsymbol{M} = [M_1, M_2, \ldots, M_N]$ by integrating the self-attention outputs $M_i$. However, these features only capture local information at the nodal level. Furthermore, it is possible for $\boldsymbol{M}$ to become an asymmetric matrix after undergoing linear transformation. To address this, we introduce second-order statistics to model feature interactions, as they have demonstrated increased sensitivity in classification [40]. These second-order statistics are computed by taking the inner product of the self-

attention output matrix, resulting in the generation of feature connections.

$$Z_{ij} = M_i^T \cdot M_j \tag{11}$$

where $Z_{ij} \in \mathbf{Z}$ and $\mathbf{Z} \in \mathbf{R}^{N \times N}$ represent the feature connection. By utilizing these features, we can generate the overall representation of BN.

In summary, the feature connection, denoted as $\mathbf{Z} = \mathbf{M}^T \cdot \mathbf{M}$, is transformed into an optimized form using the following procedure:

$$\min_{\mathbf{Z}} \|\mathbf{Z} - \mathbf{M}^T \cdot \mathbf{M}\|_F^2 \tag{12}$$

## 2.4 Introduction of regular term

Regularization refers to the introduction of constraints during the minimization of the empirical error function, which can be seen as prior information [41]. These constraints serve as guiding principles. While optimizing the error function, we tend to select the direction of gradient reduction that satisfies these constraints, leading to a solution that aligns with the prior information. This regularization approach also addresses the challenges posed by inverse problems. The resulting solution is not only existent but also unique and data-dependent. Moreover, the impact of noise on the solution is weakened, preventing overfitting. With an appropriate choice of prior, regularization ensures that even when the training set contains uncorrelated samples, the solution tends to align with the true solution (without overfitting).

In the context of BN, we can describe the attribute relationship using a hypergraph $\mathbf{G}(\mathbf{V}, \mathbf{E}, \mathbf{W})$, where $\mathbf{V}$ represents the set of nodes, $\mathbf{E}$ represents the set of hyperedges, and $\mathbf{W}$ represents the weights assigned to each hyperedge [42]. In the hypergraph $\mathbf{G}$, each brain region corresponds to a node $v \in \mathbf{V}$, and each hyperedge $e \in \mathbf{E}$ contains more than two nodes, representing the simultaneous interaction among multiple brain regions. We introduce $\mathbf{C} \in \mathbf{R}^{|V| \times |E|}$ as the point-edge association matrix of the hypergraph $\mathbf{G}$, where the values of its elements can be expressed as:

$$c(v, e) = \begin{cases} 1, & v \in e \\ 0, & v \notin e \end{cases} \tag{13}$$

The point-edge association matrix $\mathbf{C}$ allows us to calculate the node degree $\eta(v_i)$ for each node and the edge degree $\gamma(e_i)$ for each hyperedge. These can be expressed as follows:

$$\eta(v_i) = \sum_{e_i \in E} w(e_i) c(v_i, e_i), \ i = 1, ..., m \tag{14}$$

$$\gamma(e_i) = \sum_{v_i \in V} c(v_i, e_i), i = 1, ..., m \tag{15}$$

where $v_i$ represents the $i$-th node, $w(e_i)$ denotes the weight of the hyperedge $e_i$, and $m$ represents the total number of hyperedges.

The Laplacian matrix of the hypergraph provides insights into the higher-order relationships among nodes. It can be mathematically expressed as:

$$\mathbf{L}^c = \mathbf{X}_v - \mathbf{Y} \tag{16}$$

where $\mathbf{X}_v$ represents the node degree matrix with the diagonal elements denoting the $\eta(\mathbf{v}_i)$, $\mathbf{Y} = \mathbf{C}\mathbf{W}\mathbf{X}_e^{-1}\mathbf{C}^T$ is the adjacency matrix of the hypergraph, $\mathbf{W}$ is a diagonal matrix with the

hyperedge weights, and $X_e$ is the hyperedge matrix with the diagonal elements representing the $\gamma(e_i)$.

The normalized Laplacian matrix of hypergraphs can be mathematically expressed as follows:

$$L^c = I - X_v^{-\frac{1}{2}} CW X_e^{-1} C^T X_v^{-\frac{1}{2}} \tag{17}$$

where $I$ represents the identity matrix, $X_v$ is the hypergraph node degree matrix, $X_e$ is the hypergraph hyperedge degree matrix, $W$ is the diagonal matrix whose diagonal elements are the hyperedge weights, and $C \in R^{|V| \times |E|}$ is the point-edge association matrix of the hypergraph $G$.

In this study, the $k$-nearest neighbor algorithm (KNN) [43] was employed to construct hypergraphs based on the feature connections derived from Eq (12). The node degree and hyperedge degree of the hypergraph were computed, and the manifold regular term of the hypergraph was obtained, along with the addition of an $L_1$ norm regularization term. The $L_1$ norm regularization term promotes sparsity in the FBN, encouraging certain weights to approach zero. HMR helps to enhance the structured learning of feature weights. Moreover, the study utilized a multi-head attention mechanism to construct the FBN by integrating three modes: brain function network, white matter structure network, and gray matter structure network. The objective function can be formulated as follows:

$$\min_{\mathbf{Z}} \|\mathbf{Z} - \mathbf{M}^T \cdot \mathbf{M}\|_F^2 + \lambda \|\mathbf{Z}\|_1 + \beta tr(\mathbf{Z}^T \mathbf{L}^c \mathbf{Z}) \tag{18}$$

where $Z$ denotes the hypergraph representation of the FBN. The regularization parameters for the $L_1$ norm regularization term and the HMR are denoted by $\lambda$ and $\beta$, respectively. Additionally, $L^c$ corresponds to the normalized Laplacian matrix of the hypergraph.

# 3 Results

In the multi-head self-attention mechanism, we utilize the modified linear unit (ReLU) as the activation function. To mitigate the impact of different extracted features on FBN verification, we employ network edge weights as features [44]. In order to address feature extraction ambiguities, we adopt the simplest $t$-test method for feature selection. As linear kernel SVM offers simplicity and facilitates experimental result comparison, it is widely applied in small sample classification [45]. Consequently, we utilize a linear kernel SVM for classification purposes. Given the limited number of ESRDaMCI samples, we apply the leave-one-out cross-validation method to evaluate the proposed method's performance [46]. We conduct five repeated experiments to obtain an average value. Four performance metrics, namely classification accuracy (ACC), area under the ROC curve (AUC), sensitivity (SEN), and specificity (SPE), are utilized to assess the classification performance [47]. In this study, ESRDaMCI subjects are considered positive samples, whereas normal subjects are regarded as negative samples. To verify the effectiveness of FBN, we initially explore the impact of various parameters on ESRDaMCI classification. Subsequently, we determine the optimal parameters and compare them with other multi-modal BN construction methods to assess performance.

## 3.1 Parameter selection

We conducted an extensive parameter search to optimize our model. Specifically, we varied the number of heads $X$ in the multi-head self-attention mechanism, considering values such as 1, 2, 3, 5, 6, 9, 10, 15, 18, 30, 45, and 90 (factors of 90). Additionally, we explored the range of $k$, the number of neighbors, which was set between 1 and 20. The regularization parameters $\lambda$

and $\beta$ were also selected from the range of $2^{-5}$ to $2^5$. To perform feature selection, we employed the *t*-test method with a significance level of 0.05. For classification, we utilized a linear kernel SVM classifier with the parameter C set at 1. Due to the multiple parameters involved, a mesh search method was not directly applicable for finding optimal parameter values. Instead, we adopted a step-by-step approach using leave-one-out cross-validation. We determined the number of heads $X$ in the multi-head self-attention mechanism, then constructed a hypergraph based on the resulting feature connections to determine the number of neighbors $k$, and finally determined the regularization parameters $\lambda$ and $\beta$. During training, we used the ESR-DaMCI training set to find the optimal classification model, which we subsequently evaluated using the test set. The average performance across multiple tests was used to assess the model's effectiveness. We repeated the training process to identify the optimal hyperparameters, and then applied the optimal hyperparameter model to the original sample for testing purposes.

The number of multi-head attention mechanisms is a crucial parameter in FBN analysis. Increasing the number of attention heads can enhance the model's feature extraction, expression, and generalization capabilities, ultimately improving its performance and interpretability [48]. With more attention heads, the model can observe data from various perspectives and scales, thereby enhancing its ability to learn complex patterns and rules. In our study, we trained the model using different numbers of attention heads. Through leave-one-out cross-validation, we evaluated the classification performance under each configuration and compared the performance differences across different numbers of attention heads. This analysis helped us determine the optimal number of attention heads. The average ACC results are presented in Fig 2.

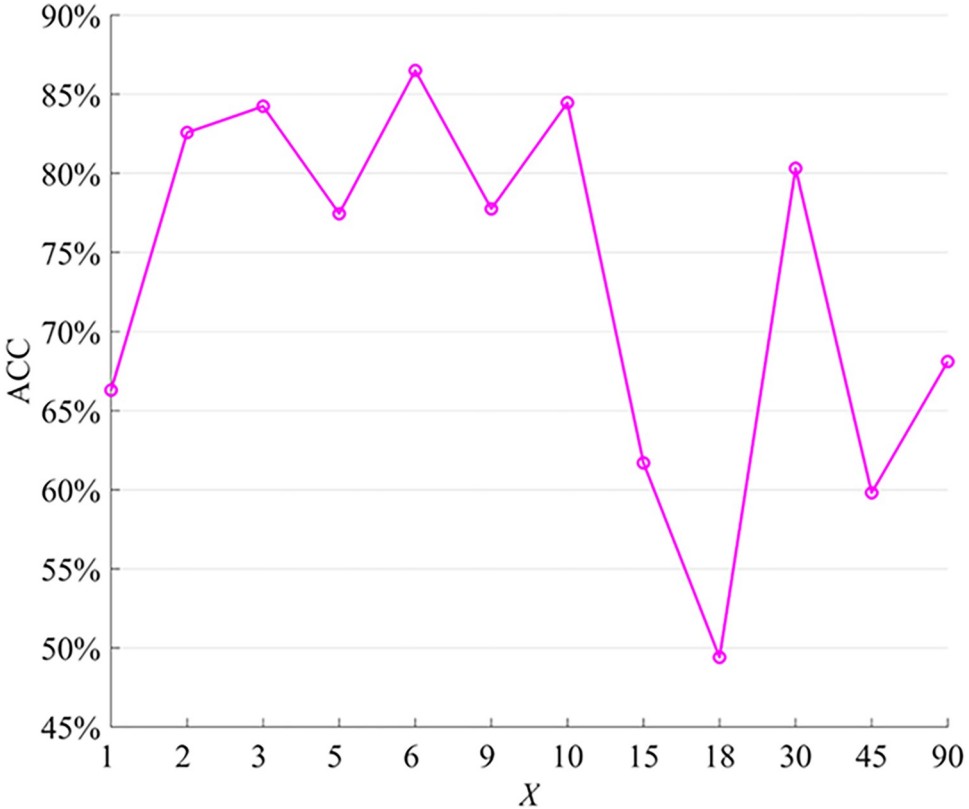

**Fig 2. Classification accuracy of different head numbers.**

Fig 2 demonstrates the significant impact of the number of attention heads on the model's performance. Generally, the overall trend of the ACC is an initial increase followed by a decrease as the number of attention heads increases. Notably, the ACC reaches its peak at approximately 86.5008% and 84.466% when the number of heads is set to 6 and 10, respectively. This improvement can be attributed to the model's ability to focus on various feature combinations and relationships simultaneously by leveraging multiple attention heads. Each attention head can effectively capture distinct local patterns and crucial information in the data, thus enhancing the model's expressiveness and performance in classification tasks. However, beyond 10 attention heads, the ACC begins to decline, with the amplitude gradually increasing. Although increasing the number of attention heads can enhance model performance, this relationship is not linear, and an excessive number of attention heads can lead to performance degradation. Notably, when the number of attention heads exceeds 10, the ACC decreases significantly to 49.3967% and 59.8039% at 18 and 45 heads, respectively. This decline may be attributed to heightened model complexity and the limited size of the training dataset, potentially resulting in overfitting or convergence difficulties. Consequently, selecting an appropriate number of attention heads is crucial for optimizing model performance.

We utilize the KNN algorithm to construct hypergraphs based on the generated feature connections. The selection of the optimal number of neighbors, denoted as $k$, plays a crucial role in determining the classification performance. It is important to strike a balance with the choice of $k$, as selecting a high value can result in overfitting, while a low value may lead to underfitting. In our study, we examine the average ACC across different neighbor numbers, as illustrated in Fig 3.

When comparing the average ACC values of ESRDaMCI subjects under different $k$ values, it is evident that ACC does not exhibit a monotonic relationship with the increase in $k$. This non-monotonic behavior can be attributed to the characteristics of the dataset and the underlying mechanics of the KNN algorithm. For instance, when $k = 1$, the ACC is 70.0586%. This relatively low ACC could be attributed to the small $k$ value being sensitive to noise and local extreme values, leading to less reliable classification results. However, as we increase $k$ to 2, the ACC improves to 75.9841%, indicating that considering more neighbors contributes to more

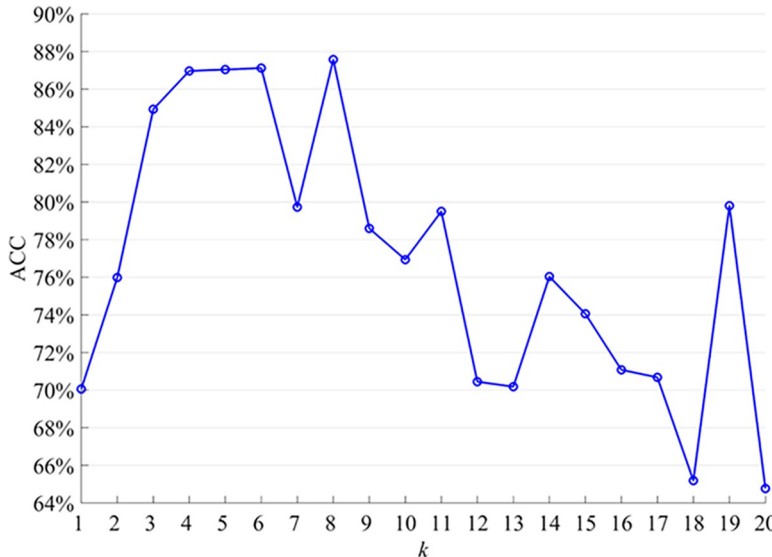

**Fig 3. Classification accuracy of different neighbors.**

robust classification outcomes. As we continue to increase the value of $k$, the ACC further improves. At $k = 4$, the ACC reaches 86.9683%, and at $k = 5$, it rises even higher to 87.0437%. The optimal ACC, peaking at 87.5716%, is achieved when $k = 8$. This improvement can be attributed to the model incorporating a larger number of neighboring nodes, thereby enhancing the accuracy of classification results. However, beyond this optimum, increasing $k$ leads to a decline in ACC, albeit at a generally high level. Notably, for specific $k$ values such as 12 and 13, the ACC experiences a significant drop to lower levels. This decline may be attributed to an excessively large $k$ value, leading to a complex node connection pattern within the hypergraph, involving numerous subgraph structures. Consequently, the model becomes more susceptible to noise and unrelated data, resulting in a decline in ACC.

The proximal operator [49] was employed to optimize and solve Eq (18). Since the $L_1$ norm regularization term is non-differentiable, the gradient descent optimization algorithm was utilized to update the feature connection $Z$ iteratively for $n$ times, with a step size $\alpha_n$. Subsequently, the soft threshold operation was applied to the elements in $Z$ using the nearest neighbor operator of the $L_1$ norm regularization term. After each iteration of gradient descent calculation, the nearest neighbor operator method was used to update $Z$ in the subsequent iteration. This process was repeated until Eq (18) converged, yielding the optimal solution for $Z$, thereby approaching the desired output, denoted as FBN. The comparison of the average ACC values of ESRDaMCI under different regularization parameters is depicted in Fig 4.

From Fig 4, we can observe that the ACC is significantly affected by different values of $\lambda$ and $\beta$. As $\lambda$ increases, the ACC initially improves and then starts to decline. This indicates that within a certain range, increasing the $L_1$ norm regularization parameter $\lambda$ enhances the

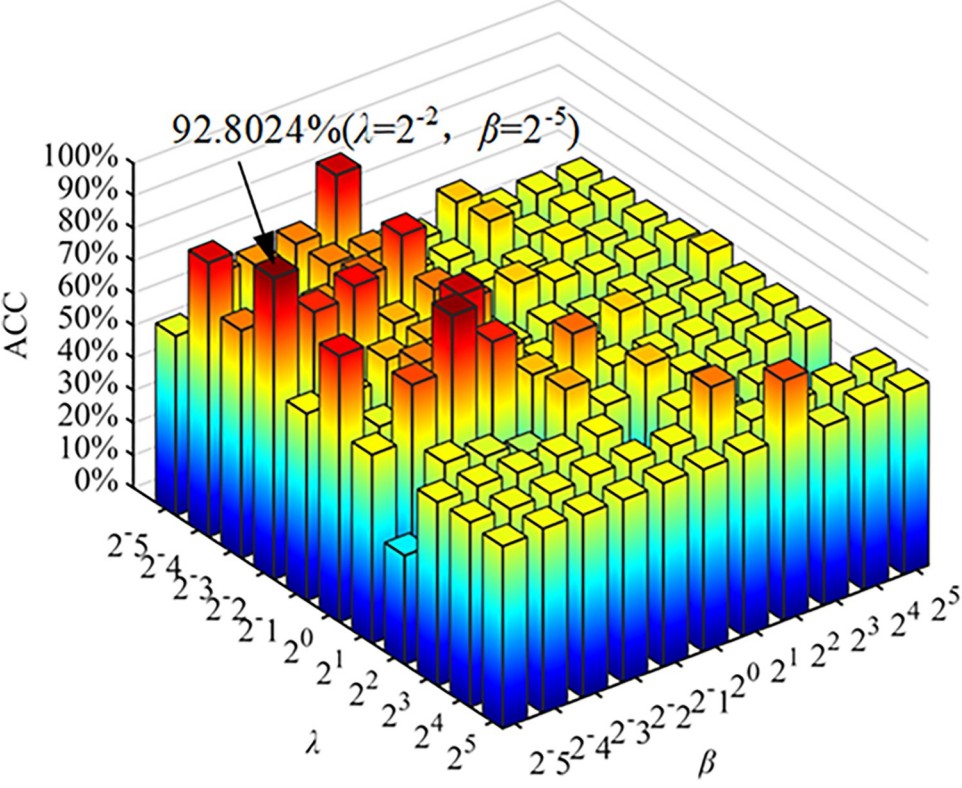

**Fig 4. Classification accuracy of different regularization parameters.** The best parameter combination is $\lambda = 2^{-2}$, $\beta = 2^{-5}$, and the corresponding ACC is 92.8024%.

model's performance. However, if $\lambda$ continues to increase, the model's performance begins to deteriorate due to overemphasis on regularization, resulting in underfitting and limiting its learning capacity. Similarly, increasing the HMR parameter $\beta$ also follows a trend of initially improving and then declining ACC. Within a certain range, an increase in $\beta$ contributes to better model performance. However, excessive values of $\beta$ can lead to regularization penalties, negatively impacting ACC. Finding the appropriate balance between $\lambda$ and $\beta$ is crucial. Through careful observation, we determined that the optimal parameter combination for our dataset and model is $\lambda = 2^{-2}$ and $\beta = 2^{-5}$, yielding an ACC of 92.8024%. This particular combination serves as a suitable regularization parameter, effectively mitigating overfitting and enhancing the model's generalization ability.

In summary, we selected a configuration with 6 heads and 8 neighbors, while employing regularization parameters $\lambda = 2^{-2}$ and $\beta = 2^{-5}$.

## 3.2 Contrast experiment

We conducted a comparative analysis with three single-modal network methods and six other multi-modal network methods to evaluate the effectiveness of our proposed method. Among the single-modal methods, we utilized the PC coefficient between fMRI time series as the functional network, PC coefficient between FN of DKI as the white matter structure network, and Euclidean distance of KV of DKI as the gray matter structure network. For the multimodal methods, we considered joint structural functional connection (JFSC) [22], multi-kernel [20], multi-view graph convolutional network (MVGCN) [50], kernel support tensor machine (KSTM) [51], diffused convolutional neural network (DCNN) [52], and bilinear attention diffusion neural network (ADBNN) [25]. It is important to note that ADBNN is specifically designed for the fusion of two modalities. To accommodate the fusion of the functional network with both the white matter structure network and the gray matter structure network, we combined the two structural networks into a single structural network, treating the functional network as nodes and the structural network as edges. All 12 methods utilized network edge weights. Feature selection was performed using a $t$-test with a significance level of 0.05, and classification was carried out using a linear kernel SVM. The classification performance was assessed against ESRDaMCI using a leave-one-out cross-validation method. The optimal classification performance is indicated in Table 2, with the values written in bold.

**Table 2. Classification performance of different methods.**

| Methods | ACC (%) | SEN (%) | SPE (%) | AUC |
|---|---|---|---|---|
| fMRI(PC) | 72.50±3.18 | 70.59±2.19 | 64.73±1.41 | 0.7347±0.0945 |
| DKI (FN) | 70.76±4.13 | 73.85±1.32 | 56.79±3.81 | 0.6819±0.0734 |
| DKI (KV) | 69.46±3.81 | 73.13±1.06 | 57.51±3.79 | 0.7054±0.0656 |
| fMRI (PC)+ DKI (FN) | 89.17±4.38 | 84.35±3.59 | 87.48±2.72 | **0.9162±0.0336** |
| fMRI (PC)+ DKI (KV) | 86.25±3.64 | 81.97±1.72 | 88.32±2.33 | 0.8511±0.0389 |
| DKI (FN)+ DKI (KV) | 73.54±2.39 | 76.11±6.70 | 77.36±3.98 | 0.7956±0.0502 |
| JSFC | 72.16±3.35 | 78.21±4.13 | 66.75±3.48 | 0.7043±0.0534 |
| Multi-kernel | 75.41±4.43 | 72.43±4.18 | 71.37±3.51 | 0.6977±0.0342 |
| MVGCN | 85.71±5.37 | 84.64±2.37 | 80.31±4.93 | 0.8135±0.0648 |
| KSTM | 73.30±2.46 | 81.34±5.22 | 67.19±2.65 | 0.8120±0.0764 |
| DCNN | 81.52±1.75 | 76.73±0.81 | 83.82±2.14 | 0.8124±0.0095 |
| ADBNN | 87.61±2.63 | 85.27±1.68 | 81.67±2.43 | 0.8971±0.0301 |
| FBN | **92.80±2.37** | **88.12±3.15** | **89.63±1.52** | 0.9069±0.0452 |

As shown in Table 2, our FBN method demonstrates superior classification performance for ESRDaMCI compared to other methods, yielding the best classification results. Notably, the FBN method achieves the highest values for ACC, SEN, and SPE, which are 92.80% ±2.37%, 88.12%±3.15%, and 89.63%±1.52%, respectively. Moreover, the error range of the FBN method is smaller compared to other methods, indicating its stability and improved classification performance. In comparison to the suboptimal fusion method that combines fMRI (PC) and DKI (FN), the FBN method exhibits an increase of 3.63%, 3.77%, and 2.15% in mean ACC, SEN, and SPE, respectively. Additionally, we observe that the multi-modal methods outperform the single-modal methods utilizing fMRI or DKI alone, highlighting the substantial improvement achieved through the integration of multiple modalities in classification tasks. Furthermore, the higher AUC values reflect the enhanced overall classification ability of the multimodal approach. For instance, the KSTM method utilizing a multi-core tensor machine yields higher performance metrics compared to the method using only fMRI. Conversely, the JSFC method, which combines fMRI and DKI, obtains lower ACC and AUC values compared to fMRI alone. This outcome may be attributed to the simple concatenation of the two modal data, with the majority of fMRI information failing to encompass structural connectivity reconstruction. To address this limitation, we employed the ADBNN method to fuse the white matter and gray matter structure networks with fMRI by simply concatenating them. However, the resulting classification performance was unsatisfactory, exhibiting a decrease of 1.56% in ACC, 5.81% in SPE, and 0.0191 in AUC compared to the fMRI (PC)+ DKI (FN) fusion method. This discrepancy is likely due to the differing expressions of features extracted from the individual white matter and gray matter structural networks, which, when simply concatenated, introduce inconsistency and result in an ineffective representation of key information after fMRI feature fusion.

### 3.3 Discriminative brain regions

In our study, we aimed to identify potential biomarkers for ESRDaMCI diagnosis. To achieve this, we constructed the FBN based on the classification performance and utilized edge weights as features for ESRDaMCI identification. The experiment was repeated 5 times, and *t*-tests were conducted for feature selection, considering features with significant differences ($P < 0.05$). As the features selected in each cross-validation iteration varied, we determined that the most relevant features for classification were those with the highest frequency across all cross-validations. Consequently, we obtained 162 features (connections). We then analyzed the brain regions involved in these connections and identified 13 discriminative brain regions with a total frequency of over 2700 instances (6 times * 90 times of cross-validation * 5 repetitions). To visualize these discriminative regions, we utilized the BrainNet Viewer toolbox and mapped them onto the ICBMl52 template space. Fig 5 displays the resulting visualization.

In Fig 5, we observed that the discriminative brain regions selected were predominantly located in the left hemisphere. The left hemisphere, which encompasses the core language area and the default network, plays a crucial role in language-related activities. This suggests that individuals with ESRDaMCI may exhibit impairments in logical task processing, language abilities, and analytical thinking. For instance, the left middle frontal gyrus (MFG.L) [53], associated with emotion and decision-making, might be linked to emotional and cognitive flexibility deficits in ESRDaMCI subjects. The left orbitofrontal medial gyrus (ORBmid.L), involved in planning and executive function, could contribute to attention and working memory impairments, potentially contributing to cognitive decline in ESRDaMCI subjects. The left orbitofrontal gyrus (ORBinf.L), responsible for emotional regulation and social behavior, might impact emotional control and social interaction in ESRDaMCI individuals. Abnormal

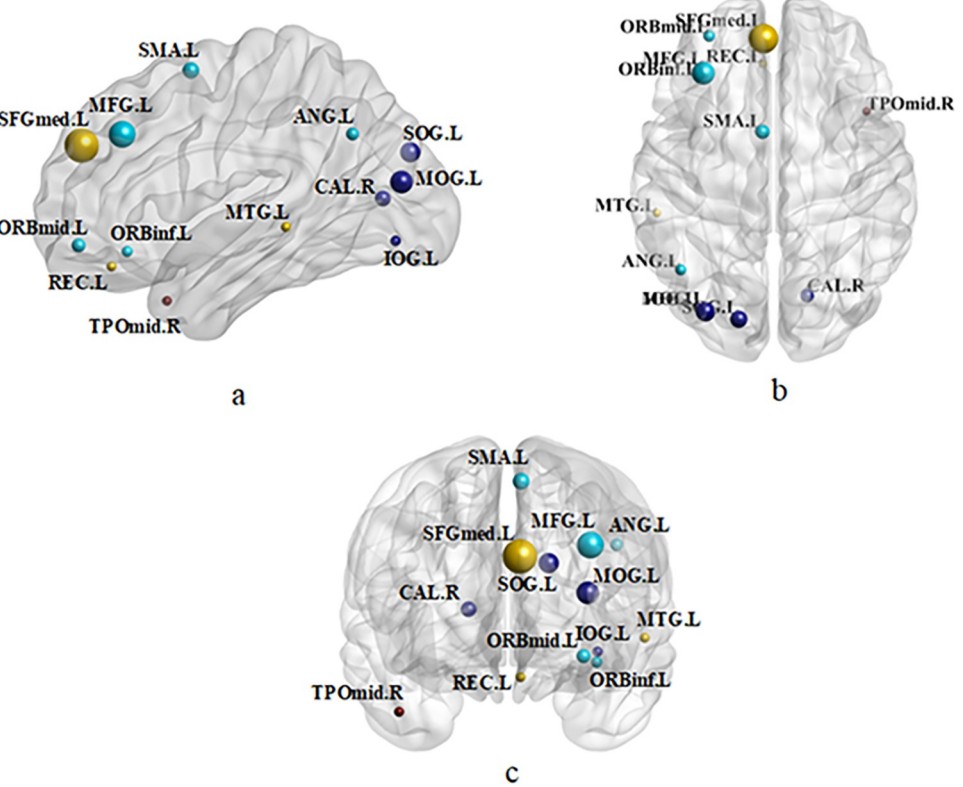

**Fig 5. Discriminative brain regions.** (a) Sagittal, (b) Axial, (c) Coronal.

or challenging muscle movement might be observed in the left supplementary motor area (SMA.L), associated with motor coordination and postural control, in ESRDaMCI subjects. The left medial superior frontal gyrus (SFMED.l), linked to working memory and thought control, could indicate a decline in cognitive flexibility and problem-solving capabilities in ESRDaMCI subjects. Language association and production might be affected by the left rectus muscle (REC.L), potentially leading to language impairments or difficulties in expression among ESRDaMCI subjects. The right peritaloid fissure cortex (CAL.R) is involved in auditory processing and spatial cognition, suggesting potential hearing impairments or difficulties with spatial orientation in ESRDaMCI individuals. The left suproccipital gyrus (SOG.L), left middle occipital gyrus (MOG.L), and left inferior occipital gyrus (IOG.L) [54] contribute to visual and spatial cognition, potentially affecting visual processing and spatial navigation abilities in ESRDaMCI subjects. The left angular gyrus (ANG.L), associated with memory and learning, might be one of the factors contributing to memory loss or learning difficulties in individuals with ESRDaMCI. The left middle temporal gyrus (MTG.L) [55], involved in language comprehension and memory, could contribute to difficulties in language comprehension and memory decline among ESRDaMCI subjects. The right temporal pole: middle temporal gyrus (TPOmid.R), responsible for emotional processing and social behavior, might influence emotional regulation and social interaction in ESRDaMCI individuals. These findings indicate that ESRDaMCI subjects exhibit alterations in memory, language, spatial visual processing, and other cognitive aspects compared to normal individuals. Some selected regions, such as the left middle temporal gyrus (MTG.L) and right temporal pole: middle temporal gyrus (TPOmid.R), align with the brain regions exhibiting significant differences in topological attributes between ESRD and normal subjects, as reported by Wu et al. [56].

**Table 3. Average topological attributes of discriminative brain regions.**

| Topo- attributes | ESRDaMCI group | Normal group | t | P |
|---|---|---|---|---|
| Degree | 45.566±2.550 | 46.243±2.313 | 3.412 | 0.001 |
| Node efficiency | 0.164±0.012 | 0.169±0.013 | 7.615 | 0.000 |
| Shortest path length | 15.648±0.015 | 15.775±0.015 | 7.046 | 0.000 |
| Clustering coefficient | 0.067±0.007 | 0.070±0.006 | 5.821 | 0.000 |
| Betweenness centrality | 3820.110±544.912 | 3787.884±522.801 | -2.047 | 0.004 |
| Small-world attribute | 0.589±0.085 | 0.664±0.037 | 2.458 | 0.003 |

We conducted an analysis of various topological attributes, including degree, node efficiency, shortest path length, clustering coefficient, betweenness centrality and small-world attribute of the identified discriminative brain regions. A comparative analysis between the two groups was performed using independent sample $t$-test, with statistical significance set at $P < 0.05$. The average topological properties of these discriminative brain regions are summarized in Table 3, a significance level of $P < 0.05$ was used to determine statistical significance. Notably, all the computed $P$-values were found to be less than 0.05, indicating statistical significance. These results affirm the accuracy of our selection of discriminative brain regions associated with ESRDaMCI.

The efficiency of BN nodes reflects the efficiency of information transmission between different brain regions. Jiang et al.'s research [57] indicates that BN node efficiency exhibits higher sensitivity to scale scores. In our study, we utilized PC and two-tailed significance test to examine the correlation between node efficiency and MoCA scale scores within the ESRDaMCI group. This analysis aimed to explore the relationship between cognitive function and network connectivity characteristics. The correlation coefficient $r$ and significance level $P$ are presented in Table 4, with brain regions demonstrating significant correlations marked with an asterisk (*), a significance level of $P < 0.05$ was used to determine statistical significance. Table 4 reveals that among the discriminative brain regions, six regions exhibited notable correlations, namely the left middle frontal gyrus (MFG.L), left orbital middle frontal gyrus (ORBmid.L), left supplementary motor area (SMA.L), left medial superior frontal gyrus (SFMED.L), left rectus muscle (REC.L), and right temporal pole: middle temporal gyrus (TPOmid.R). The nodal efficiency of these regions displayed a negative correlation with the MoCA scale score. Specifically, individuals with lower MoCA scale scores tended to exhibit reduced BN node

**Table 4. Correlation between node efficiency of discriminative brain regions and scale scores.**

| AAL number | Regions | Node efficiency | r | P |
|---|---|---|---|---|
| 7 | MFG.L* | 0.166±0.078 | -0.288 | 0.037 |
| 9 | ORBmid.L* | 0.161±0.084 | -0.245 | 0.043 |
| 15 | ORBinf.L | 0.160±0.055 | 0.086 | 0.246 |
| 19 | SMA.L* | 0.152±0.093 | -0.232 | 0.046 |
| 23 | SFGmed.L* | 0.166±0.081 | -0.304 | 0.021 |
| 27 | REC.L* | 0.172±0.065 | -0.277 | 0.039 |
| 44 | CAL.R | 0.181±0.065 | 0.090 | 0.606 |
| 49 | SOG.L | 0.184±0.043 | -0.153 | 0.133 |
| 51 | MOG.L | 0.175±0.045 | -0.047 | 0.228 |
| 53 | IOG.L | 0.154±0.041 | 0.050 | 0.579 |
| 65 | ANG.L | 0.156±0.047 | 0.080 | 0.355 |
| 85 | MTG.L | 0.151±0.067 | 0.053 | 0.400 |
| 88 | TPOmid.R* | 0.150±0.044 | -0.290 | 0.032 |

efficiency. These findings suggest that the decline in cognitive function observed in ESR-DaMCI may be associated with abnormal connectivity and degeneration in these specific brain regions. It is plausible that degenerative changes have occurred in these six brain regions among individuals with ESRDaMCI.

## 4 Discussion

In recent years, there has been a growing interest among researchers in exploring various aspects of MCI, including its epidemiology, clinical features, neuroimaging, biomarkers, disease mechanisms, neuropathology, and clinical trials. However, determining the threshold for diagnosing MCI in conjunction with other diseases remains a challenge. Specifically, limited research has been conducted on ESRDaMCI. Most existing BN fusion methods focus on extracting interaction features separately from pairs of network nodes within each mode. These methods then combine these feature vectors from multiple groups, overlooking the potential correlation between different modal features and valuable information that may exist among more than two brain regions during the feature extraction process. To tackle this limitation, we propose a novel FBN model that integrates three modes of BN using a multi-head self-attention mechanism. This model is applied to the classification of ESRDaMCI. Compared to conventional fusion networks, our proposed model's self-attention mechanism can automatically determine the importance of each modal network in classification tasks by learning the interactions among the brain function network, white matter structure network, and gray matter structure network during the fusion process. This enables better utilization of information contained within multimodal data, thus improving the performance and expressive capabilities of the fusion network. Notably, fusing information from multiple modes leads to higher ACC compared to using a single mode exclusively. It is worth highlighting that multimodal BN analysis differs significantly from traditional multimodal data analysis due to the natural connection between structural and functional networks. The structure of the brain often influences its function, thereby allowing for further exploration of the associations between multiple modalities.

Several studies have focused on BN modeling for multimodal fusion. For instance, Misic et al. [58] utilized the partial least squares method to investigate the relationship between structural and functional networks, identifying closely related subnetworks and important nodes. Goni et al. [59] directly predicted functional connections using structural connection strength and observed high consistency in certain brain regions. However, these methods simply fuse single-modal data without considering how to integrate multiple modal information effectively. It is crucial to accurately describe the interplay between brain regions. In our approach, instead of directly using or concatenating individual networks, we employ a three-mode network to automatically learn interactions between nodes. By constructing a hypergraph based on the generated feature connections, multiple nodes are grouped within the same hyperedge, effectively capturing potential correlation information across multiple modes. Moreover, the incorporation of an $L_1$ norm regularization term facilitates feature sparsity and accuracy by eliminating redundant features. The inclusion of the HMR preserves discriminative information from each subject and induces more distinctive features. By adjusting the complexity of the FBN model, we enhance its generalization ability and stability.

In this study, we observed significant differences between the ESRDaMCI and Normal groups in terms of discriminative brain regions. Specifically, the ESRDaMCI group exhibited lower values in degree, node efficiency, clustering coefficient, small-world attribute, and shortest path length, indicating weakened connectivity, reduced transmission efficiency, and poorer information processing abilities compared to the Normal group. Additionally, the betweenness

centrality was significantly higher in the ESRDaMCI group. These findings suggest that the weakened connectivity and reduced transmission efficiency of discriminative brain regions may contribute to the mechanisms underlying ESRDaMCI. Furthermore, we identified several discriminative brain regions in the ESRDaMCI group, including the left middle frontal gyrus (MFG.L), left orbital middle frontal gyrus (ORBmid.L), left supplementary motor area (SMA.L), left medial superior frontal gyrus (SFMED.L), left rectus (REC.L), and right temporal pole: middle temporal gyrus (TPOmid.R). Notably, the node efficiency of them exhibited a negative correlation with the MoCA scale scores. Node efficiency, which measures the information transfer efficiency of specific brain regions in the BN, demonstrated a negative correlation with cognitive impairment as assessed by the MoCA scale. This suggests that brain regions with lower node efficiency may be associated with the degree of cognitive impairment and potentially play a significant role in ESRDaMCI. Consequently, ESRDaMCI requires increased attention in clinical practice, emphasizing the importance of screening, early diagnosis, and early treatment to prevent or delay disease progression.

However, it is important to acknowledge the limitations of our study. Firstly, the fiber bundle tracking algorithm we employed, known as FACT, relies on continuous tracking along the primary direction of diffusion. While this method is computationally efficient and yields prompt results, it does not account for the uncertainty of fiber path direction or the presence of cross fibers. In the future, it would be beneficial to explore the use of probabilistic fiber bundle tracking techniques. Additionally, in our study, we solely utilized attentional mechanisms to establish feature connections. Considering that fMRI data captures temporal changes through fluctuations, it is crucial to incorporate temporal information into the representation learning process. Moving forward, we intend to investigate the combined utilization of temporal and spatial attention mechanisms to characterize both temporal and spatial features of the FBN. Another limitation is the limited number of samples used in our study, focusing solely on binary classification. To validate our approach more comprehensively, it is necessary to collect a larger sample size and establish multiple classification tasks. For example, including ESRD subjects without cognitive impairment to create a three-classification problem would provide additional insights. Furthermore, the brain regions in our study were defined using the AAL template, with a selection of only 90 regions (excluding the cerebellum) to construct the FBN. In future studies, alternative templates that divide the brain into finer regions could be considered to further refine the analysis.

## 5 Conclusion

In this study, we propose a multi-head self-attention mechanism model to integrate brain functional networks, white matter structural networks, and gray matter structural networks, constructing the FBN for a comprehensive understanding of the relationship between cognitive function and network connectivity characteristics in ESRDaMCI. This model effectively combines DKI and fMRI data, offering a novel approach to the analysis of BN. We demonstrate its efficacy on real ESRDaMCI datasets, achieving good performance. Furthermore, by selecting informative features, we identify discriminative brain regions and analyze their topological attributes through statistical analysis. This validates the effectiveness of our method and identifies brain regions that exhibit significant correlations with MoCA scale scores, providing insights into the underlying mechanisms of ESRDaMCI.

## Acknowledgments

Thanks to Jiaxing Yang and Yutao Zhang for their support and feedback on the earlier version of this study, and great help in the processing and correction of image data.

## Author Contributions

**Data curation:** Tongqiang Liu.

**Formal analysis:** Jie Chen, Haifeng Shi.

**Funding acquisition:** Haifeng Shi.

**Methodology:** Jie Chen, Tongqiang Liu.

**Writing – original draft:** Jie Chen.

**Writing – review & editing:** Tongqiang Liu, Haifeng Shi.

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
