## [Decision Letter · Decision Letter 0]

25 Mar 2024

PONE-D-24-00628End-stage renal disease accompanied by mild cognitive impairment: A study and analysis of trimodal brain network fusionPLOS ONE

Dear Dr. Shi,

Thank you for submitting your manuscript to PLOS ONE. After careful consideration, we feel that it has merit but does not fully meet PLOS ONE’s publication criteria as it currently stands. Therefore, we invite you to submit a revised version of the manuscript that addresses the points raised during the review process.

We look forward to receiving your revised manuscript.

Kind regards,

Cota Navin Gupta

Academic Editor

PLOS ONE

Journal Requirements:

"Top Talent of Changzhou “The 14th Five-Year Plan” High-Level Health Talents Training Project (grant No. 2022CZBJ072)."

4. In this instance it seems there may be acceptable restrictions in place that prevent the public sharing of your minimal data. However, in line with our goal of ensuring long-term data availability to all interested researchers, PLOS’ Data Policy states that authors cannot be the sole named individuals responsible for ensuring data access (http://journals.plos.org/plosone/s/data-availability#loc-acceptable-data-sharing-methods).

**Additional Editor Comments:**

Please address the below queries

Major comments

1)In introduction you state “Investigating the relationship between changes in brain functional networks and MCI in patients with ESRD is an area of active research interest [17]” …Reference [17] has no mention of ESRD?

2)You introduce “mode of brain network” with this sentence “The utilization of multimodal fusion methods can offer more comprehensive and accurate brain information, revealing correlations between different modes of BN” what does mode mean here

3)Add a citation for Montreal Cognitive Assessment (MoCA)..Also typo in reference 27.

4)What is b0 image? what is FN? What is HMR?

5)Reference for “A deterministic fiber bundle tracking algorithm was employed, with tracking parameters set as anisotropy index < 0.15 and tracking angle < 45° to calculate indicators such as FN between brain regions”

6)Section 2.3.3, references missing for first 6 lines.

7)C is correlation matrix and also point edge association matrix?

8)Figures are of poor quality...so please include better quality ones

Minor comments

1)Check English in below sentences.

a) It offers a powerful representation of the patterns of interaction between different regions of brain.

b) We are able to obtain information that can identify individual participants.

c)Numerous other typos and grammatical errors also exists. please check the entire paper.

Reviewers' comments:

Reviewer's Responses to Questions

**Comments to the Author**

1. Is the manuscript technically sound, and do the data support the conclusions?

Reviewer #1: Yes

Reviewer #2: Yes

Reviewer #3: Yes

2. Has the statistical analysis been performed appropriately and rigorously? 

Reviewer #1: Yes

Reviewer #2: Yes

Reviewer #3: Yes

3. Have the authors made all data underlying the findings in their manuscript fully available?

Reviewer #1: Yes

Reviewer #2: Yes

Reviewer #3: Yes

4. Is the manuscript presented in an intelligible fashion and written in standard English?

Reviewer #1: Yes

Reviewer #2: Yes

Reviewer #3: Yes

5. Review Comments to the Author

Reviewer #1: 1. Compared with the existing state-of-the-art methods, what are the core innovations in your work?

2. The Abstract should be more concise.

3. There are too many processes and methods introduced in section 2.1 and 2.2, and these are not your main work.

4. The attached figures look a little blurry and needs to be replaced with a clearer version.

Reviewer #2: In this research paper, the authors delve into the examination of alterations in individuals grappling with end-stage renal disease, especially those presenting with mild cognitive impairment. They introduce a model featuring a multi-head self-attention mechanism to amalgamate brain functional networks, white matter structural networks, and gray matter structural networks. This integration culminates in the formation of brain fusion networks.

The study is potentially interesting, the methods are sounding to me, the work seems well performed and sufficiently detailed, the figures and tables are clear.

Some points remain to be clarified:

Reference for sentence “Epidemiological surveys indicate that cognitive dysfunction is highly prevalent among ESRD patients, with conservative estimates suggesting that 16% to 38% of them experience cognitive impairment.”

In my opinion the sentence “Mild cognitive impairment (MCI) serves as an intermediate stage between normal cognitive function and cognitive dysfunction.” could be improved to “Mild cognitive impairment (MCI) represents the intermediate stage between the normal decline in memory and thinking associated with aging and the more pronounced cognitive deterioration observed in dementia.

Add references in the sentence “These techniques include functional magnetic resonance imaging (fMRI) [6], structural magnetic resonance imaging [7], magnetic resonance spectroscopy [8], fluorodeoxyglucose positron emission tomography [9], single photon emission tomography [10], arterial spin labeling [11], diffusion tensor imaging [12], and diffusion kurtosis imaging (DKI) [13]”

The sample sizes of the two groups are not well balanced. There are much fewer controls than MCI subjects.

In the discussion the authors should add the strengths.

Reviewer #3: This manuscript utilized a multi-head self-attention mechanism to fuse brain functional networks, white matter structural networks, and gray matter structural networks, and constructed brain fusion networks (FBN). FBN is employed in the diagnosis and prediction of ESRDaMCI to evaluate its classification performance and investigate the correlation between discriminative brain regions and cognitive dysfunction. This method identifies brain regions that exhibit significant correlations with MoCA scale scores, providing insights into the underlying mechanisms of ESRDaMCI. The content of the manuscript is well-organized, the methods they used is novelty. There are some minor problems need to be addressed:

1.The title of the figures should not appeared separately in the manuscript.

2.Figure legends should be added and listed at the end of the References.

6. PLOS authors have the option to publish the peer review history of their article (what does this mean?). If published, this will include your full peer review and any attached files.

Reviewer #1: No

Reviewer #2: **Yes: **Daniele Corbo

Reviewer #3: No

---

## [Author Response · Author response to Decision Letter 0]

6 May 2024

Journal Requirements:

Reply: It has been checked and corrected.

Reply: We can provide it if necessary.

"Top Talent of Changzhou “The 14th Five-Year Plan” High-Level Health Talents Training Project (grant No. 2022CZBJ072)."

Reply: Funders Haifeng Shi: Topic selection design, paper review.( As explained in the cover letter)

4. In this instance it seems there may be acceptable restrictions in place that prevent the public sharing of your minimal data. However, in line with our goal of ensuring long-term data availability to all interested researchers, PLOS’ Data Policy states that authors cannot be the sole named individuals responsible for ensuring data access (http://journals.plos.org/plosone/s/data-availability#loc-acceptable-data-sharing-methods).

Reply: Contact information for the Ethics Committee: 2392058632@qq.com/15804942480

Reply: It has been checked and modified.

Additional Editor Comments:

1)In introduction you state “Investigating the relationship between changes in brain functional networks and MCI in patients with ESRD is an area of active research interest [17]” …Reference [17] has no mention of ESRD?

Reply: Reference [17] has been replaced.

2)You introduce “mode of brain network” with this sentence “The utilization of multimodal fusion methods can offer more comprehensive and accurate brain information, revealing correlations between different modes of BN” what does mode mean here?

Reply: The mode here refers to the different modes of brain networks constructed, including functional networks, white matter structural networks and gray matter structural networks.

3)Add a citation for Montreal Cognitive Assessment (MoCA)..Also typo in reference 27.

Reply: Reference 27 label has been moved to Montreal Cognitive Assessment (MoCA).

4)What is b0 image? what is FN? What is HMR?

Reply: b0 image: A ‘b0 image’ is an image taken when no diffusion-weighted gradient is applied, also known as a baseline image or an image without diffusion-weighted.

FN: The number of fiber bundles.

HMR: The hypergraph manifold regularization term.

5)Reference for “A deterministic fiber bundle tracking algorithm was employed, with tracking parameters set as anisotropy index < 0.15 and tracking angle < 45° to calculate indicators such as FN between brain regions”?

Reply: Reference has been added.

6)Section 2.3.3, references missing for first 6 lines.

Reply: A reference has been added.

7)C is correlation matrix and also point edge association matrix?

Reply: Yes, it's been changed to remove ambiguity.

8)Figures are of poor quality...so please include better quality ones

Reply: Figures that have been processed with PACE have been uploaded.

Minor comments

1)Check English in below sentences.

a) It offers a powerful representation of the patterns of interaction between different regions of brain.

Reply: "It" has been changed to "BN".

b) We are able to obtain information that can identify individual participants.

Reply: This sentence has been deleted.

c)Numerous other typos and grammatical errors also exists. please check the entire paper.

Reply: The entire paper has been checked and amended appropriately.

Review Comments to the Author

Reviewer #1:

1. Compared with the existing state-of-the-art methods, what are the core innovations in your work?

Reply: a) For the first time, an individual-based gray matter network specific to ESRDaMCI was constructed using KV extracted from DKI; b) The application of a multi-head self-attention mechanism is a novel approach to model ESRDaMCI's brain function network, as well as the fusion network of white matter structure and gray matter structure; c) By utilizing the aforementioned fusion network, this study successfully identifies discriminative brain regions associated with the pathogenesis of ESRDaMCI. Furthermore, it analyzes the changes in topological attributes of these discriminative brain regions and identifies brain regions that significantly contribute to cognitive dysfunction.

2. The Abstract should be more concise.

Reply: The Abstract has been simplified appropriately.

3. There are too many processes and methods introduced in section 2.1 and 2.2, and these are not your main work.

Reply: Parts have been deleted.

4. The attached figures look a little blurry and needs to be replaced with a clearer version.

Reply: Figures that have been processed with PACE have been uploaded.

Reviewer #2: 

1. Reference for sentence “Epidemiological surveys indicate that cognitive dysfunction is highly prevalent among ESRD patients, with conservative estimates suggesting that 16% to 38% of them experience cognitive impairment.”

Reply: It has been modified.

2. In my opinion the sentence “Mild cognitive impairment (MCI) serves as an intermediate stage between normal cognitive function and cognitive dysfunction.” could be improved to “Mild cognitive impairment (MCI) represents the intermediate stage between the normal decline in memory and thinking associated with aging and the more pronounced cognitive deterioration observed in dementia.”

Reply: It has been modified to "Mild cognitive impairment (MCI) represents the intermediate stage between the normal decline in memory and thinking associated with aging and the more pronounced cognitive deterioration observed in dementia".

3. Add references in the sentence “These techniques include functional magnetic resonance imaging (fMRI) [6], structural magnetic resonance imaging [7], magnetic resonance spectroscopy [8], fluorodeoxyglucose positron emission tomography [9], single photon emission tomography [10], arterial spin labeling [11], diffusion tensor imaging [12], and diffusion kurtosis imaging (DKI) [13]”

Reply: We already have one reference for each technique, so we shouldn't need to add any more.

4. The sample sizes of the two groups are not well balanced. There are much fewer controls than MCI subjects.

Reply: Patients with mild cognitive impairment were easier to identify and recruit into the study, while the control group was relatively difficult to recruit because they typically did not actively seek medical help or participate in relevant studies. In studies involving medical or health information, the control group had higher expectations and demands for privacy protection, which may have influenced their willingness to participate. The study focused on patients with mild cognitive impairment, so the sample size of the patient group may be relatively larger to better understand and analyze this specific group.

5.In the discussion the authors should add the strengths.

Reply: There are advantages to writing in the first paragraph of the discussion: “Compared to conventional fusion networks, our proposed model's self-attention mechanism can…thereby allowing for further exploration of the associations between multiple modalities.”

Reviewer #3: 

1.The title of the figures should not appeared separately in the manuscript.

Reply: It has been deleted.

2.Figure legends should be added and listed at the end of the References.

Reply: It has been added.

---

## [Editor Report · Decision Letter 1]

23 May 2024

End-stage renal disease accompanied by mild cognitive impairment: A study and analysis of trimodal brain network fusion

PONE-D-24-00628R1

Dear Dr. Shi,

We’re pleased to inform you that your manuscript has been judged scientifically suitable for publication and will be formally accepted for publication once it meets all outstanding technical requirements.

Kind regards,

Cota Navin Gupta

Academic Editor

PLOS ONE

Additional Editor Comments (optional):

Good work in addressing the reviewer comments
---

## [Editor Report · Acceptance letter]

4 Jun 2024

PONE-D-24-00628R1 

PLOS ONE

Dear Dr. Shi, 

I'm pleased to inform you that your manuscript has been deemed suitable for publication in PLOS ONE. Congratulations! Your manuscript is now being handed over to our production team.

Kind regards, 

on behalf of

Dr. Cota Navin Gupta 

Academic Editor

PLOS ONE